# Precipitation Strengthening of Cu–Ni–Si Alloy

**DOI:** 10.3390/ma13051182

**Published:** 2020-03-06

**Authors:** Beata Krupińska, Zbigniew Rdzawski, Mariusz Krupiński, Wojciech Pakieła

**Affiliations:** 1Department of Engineering Materials and Biomaterials, Faculty of Mechanical Engineering, Silesian University of Technology, Konarskiego St. 18a, 44-100 Gliwice, Poland; mariusz.krupinski@polsl.pl (M.K.); wojciech.pakiela@polsl.pl (W.P.); 2Łukasiewicz Research Network—Institute of Non-Ferrous Metals, Sowinskiego St. 5, 44-100 Gliwice, Poland; zbigniew.rdzawski@imn.gliwice.pl

**Keywords:** heat treatment, advanced materials characterization, rhenium modification

## Abstract

The work examines the effect of rhenium addition on the structure and properties of Cu–2Ni–1Si alloys. The aim of this work was to answer the question of how the addition of rhenium will affect the strengthening mechanisms of rhenium-modified, saturated, plastically deformed and aged Cu–2Ni–1Si alloys. How will this affect the crystallization process? What effect will it have on the properties? Scanning electron microscopy (SEM) and analysis of chemical composition in microareas (energy-dispersive X-ray spectroscopy, EDS), light microscopy, measurements of microhardness and conductivity of the alloys were used for the investigations. Research on chemical and phase composition were carried out with application of transmission electron microscopy (TEM), and scanning transmission electron microscopy (STEM). Modification with rhenium has caused an increase in hardness as a result of precipitation of small phases with rhenium. As the effect of supersaturation, cold plastic treatment as well as aging small phases with rhenium with a size of 200 nm to 600 nm causes both reinforcement of the alloy and makes recrystallization impossible. Re-addition also influences the stabilization of the structure.

## 1. Introduction

The dynamic development of modern technologies forces the development of materials engineering and continuous improvement of existing materials and the development of new materials. The requirements for the applied materials are increasingly higher and more precise. These requirements apply to the whole spectrum of properties at the same time. Modern materials are expected to be both durable and reliable during operation, and thus have good strength, electrical and thermal properties [1,2,3,4].

Advancements in the field of electrical engineering, power engineering, electronics and aeronautics require intensive research into the materials employed in these fields. Copper alloys are commonly used metal engineering materials. However, the growing range of applications has led to an expansion of the research to date, with the aim of developing alloys with better performance properties, i.e., high hardness and abrasion resistance, at the same time as electrical conductivity being maintained or slightly reduced. A significant part of these properties are met by precipitation-hardened copper alloys such as Cu–Fe, Cu–Cr, Cu–Co, Cu–Ni–Si and Cu–Ni–Si–Cr [1,2,5,6,7,8,9,10].

Cu–2Ni–1Si alloys, whose mechanical properties can be shaped by heat and plastic treatment, are increasingly becoming the subject of research. The following is often used as a modifier in investigations of Cu–2Ni–1Si alloys: Cr or Cr and Mg or Cr and Ti. In the obtained Cu–Ni–Si–Cr, Cu–Ni–Si–Cr–Mg or Cu–Ni–Si–Cr–Ti alloys, the results of investigations indicate a change in morphology, resulting in more beneficial mechanical properties and in some cases also beneficial changes in conductivity. It is known that very often during heat or plastic treatment, strength properties are enhanced, but electrical properties deteriorate [3,4,5,6,7,8,9,10,11].

In the research [1] the significant role of nanometric, coherent precipitates, created as the result of supersaturation and aging of Cu–Ni–Si alloy, has been indicated. The precipitates (Ni_2_Si) grow as time of the aging goes on, the distance between them also increases. Studies [12,13] pointed to the presence of phases β (Ni_3_Si) i δ (Ni_2_Si) in Cu–Ni–Si alloys.

Broad advances are also observed in superalloys, i.e., metal alloys, usually based on iron, nickel, molybdenum cobalt and a wide range of rare earths [14,15,16,17]. They are characterized by excellent mechanical properties even at high temperatures. Their corrosive properties are also very good, even under extreme conditions. These properties are very often obtained by modification with rhenium, which improves all the relevant properties of superalloys. It has been established as a result of the investigations carried out that this is due to inhibiting the recrystallization of the alloy grains and increasing the average grain size. The tendency of rhenium to form clusters, still at the time when they are “nano” in size, allows to inhibit the migration of other atoms without destroying the heterogeneity of the alloy at the micro level. This phenomenon is often referred to as the “rhenium effect” [18,19,20,21,22].

Current investigations are carried out due to good functional properties of rhenium at the lowest possible concentration due to its scarce resources and increasing interest in it. There is in the world 10 times less rhenium than gold. It was examined that 1 ton of the Earth’s crust contains 50 g Cu, 0.7 g Ag, only 0.1 g Au and 0.01 g Re [18,19,20,21,22,23,24,25].

The analysis and investigations of Cu–2Ni–1Si alloy modified with Cr, Ag [2] as well as the investigations and results of our own research and the development of science and increasing demands have acted as the basis for the investigation of Cu–2Ni–1Si alloys modified with rhenium. Re content was determined at the level below 1% due to scarce resources and the results obtained for Cu–2Ni–1Si–0.8Cr alloys modified with rhenium [3].

The studies have been undertaken because the literature concerning modification of Cu–2Ni–1Si alloys by Re-addition is quite poor. However rhenium properties, described in the literature, motivate its use as an alloy element at a concentration of 4% mass, which causes stabilization of the structure, blocks the recrystallization and, therefore, increases mechanical properties. In the research rhenium has been used as a modifier at a concentration of 0.5–0.6 mass% to verify whether identical mechanisms are present with smaller Re concentration.

## 2. Materials and Methods

Because of technological problems connected with Re addition to the alloy concerning high remelting temperature and therefore the possibility of copper overheating, rhenium was added in form of Ni–Re master alloy.

The investigations concerned Cu–2Ni–1Si–0.6Re alloys (Table 1). The alloys were subjected to heat treatment and plastic deformation. First, the alloys were supersaturated at 950 °C for 1 h in a protective atmosphere of argon, then they were cooled with water. The next step was plastic deformation to 50%. The alloys were aged at 450 °C for 1 h also in the atmosphere of argon. The treatment was performed using a thermoplastic working simulator DSI (Dynamic System Inc., Austin, TX, USA) Gleeble 3800. Prior to each experiment involving heat treatment or cold plastic deformation, and before introducing a protective gas into the working chamber of the Gleeble 3800 simulator, a vacuum of 0.1 mbar was first created in the chamber and then a protective gas—argon with a partial pressure of about 200 mbar—was introduced.

Research concerning changes of microstructure and being the effect of heat treatment of phases and participates have been conducted with application of the following devices: scanning electron microscope Zeiss Supra 35 device (SEM, Thornwood, NY, USA) in high-resolution mode, and a high-resolution transmission electron microscope (TEM, scanning transmission electron microscope, STEM, (FEI Company, Hillsboro, OR, USA). The cross-section has been prepared with microsampling method. To gather material to prepare lamellas (used in TEM analysis) both energy-dispersive X-ray spectroscopy (EDS) analysis and microstructures photographs were taken to allow the sample to be extracted from the place where participates of interest were contained. Subsequently, with application of SEM/XE-PFIB FEI Helios G4 PFIB CXe, lamellas of the size 5 × 5 μm were prepared.

The following investigations were performed in order to determine the relationship between the crystallization kinetics of the alloy and the chemical composition and microstructure of Cu alloy:thermo-derivative analysis using the UMSA device (Method and Apparatus for Universal Metallurgical Simulation and Analysis-Patent Serial No. PCT/CA02/01903, Silesian University of Technology, Gliwice, Poland) equipped with a computer-controlled cooling system, which allows to set flexibly the cooling rate applied of the Cu–2Ni–1Si (11.36 g) and Cu–2Ni–1Si–0.6Re (24.2 g) alloys. The samples for thermo-derivative analysis were prepared with a diameter of Ø8 mm and a height of 10 mm. Holes were also made for thermocouples in the samples where the thermal node occurs for this type and arrangement of the sample geometry. K-type thermocouples were used for testing due to their linear measurement characteristics;the supersaturation temperature of Cu–2Ni–1Si and Cu–2Ni–1Si–0.6Re alloys was 950 °C, heating time 1 h (in a protective atmosphere of argon) and subsequent ageing at a temperature of 450 °C (in a protective atmosphere of argon); in the investigation the simulator of heat and plastic treatment used was a DSI (Dynamic System Inc., Austin, TX, USA) Gleeble 3800;alloy structure was examined using electron scanning microscope using the Zeiss Supra 25 device (SEM, Thornwood, New York, USA) within the high-resolution mode;microstructure and chemical composition investigations using EDS microanalysis was undertaken on the scanning electron microscope Zeiss Supra 25;microstructure, chemical and phase composition were carried out using a FEI TITAN TEM, using selected area diffraction (SAD, FEI Company, Hillsboro, OR, USA) at 300 kV acceleration voltage, to identify the crystalline phase structures. The obtained diffraction patterns were analysed by specialized software dedicated to solving electron diffraction patterns;the microhardness test was performed on the hardness tester Vickers FUTURE-TECH (FM-ARS9000, Future-Tech, Tokyo, Japan) with 1000 gf for 15 s;the measurement of electrical conductivity was performed with the Sigmatest Foerster device (FOERSTER, Pittsburgh, PA, USA);

The samples prepared for observation of the microstructure were ground and polished mechanically, and then etched electrochemically in an electrochemical reagent consisting of: ethanol, phosphoric acid, propanol, urea and distilled water, or in a reagent with the following composition: iron chloride, hydrochloric acid and ethyl alcohol.

## 3. Results and Discussion

The paper presents how the addition of Re and the following changes in the cooling rate of the alloy (changes of sample mass and its heat capacity) affect the kinetics of crystallization.

Results of the applied thermo-derivative analysis have been shown on the charts for Cu–2Ni–1Si alloy in Figure 1 (Table 2) and for Cu–2Ni–1Si–0.6Re alloy in Figure 2 (Table 2).

The Cu–2Ni–1Si alloy crystallizes at point I at the temperature T_L_ = 1098 °C and ends crystallization at point II at the temperature T_SOL_ = 1057 °C. There is no clear bend on the derivation curve indicating the crystallization of other phases and eutectics. (Figure 1, Table 2) However, Ni phases are visible at the phase α boundary in the images of the microstructure (Figure 3 and Figure 4). The addition of rhenium in the Cu–2Ni–1Si alloy reduces both the temperature at point I, which is T_L_ = 1083 °C, and the temperature at point II, T_SOL_ = 1014 °C, causing extension of the crystallization time (Figure 2, Table 2). There is no clear bend on the derivation curve (Figure 2) of Cu–2Ni–1Si–0.6Re, either.

Cooling rate was calculated as Equation (1):(1)$$CR=\frac{T_{L}-T_{SOL}}{t_{SOL}-t_{L}}$$

Microstructure and microanalysis EDS results of Cu–2Ni–1Si alloy are presented in Figure 3 and Figure 4. In Figure 3, the more fragmented microstructure (grain size of approximately 50–150 μm) with intermetallic phases including β i δ on the α phase boundary can be seen. However Re addition causes the structure to be more coarse (grain size of approximately 125–320 μm) and participates of Ni_2_Si phases are also on the α phase boundary (Figure 5).

However, also in this case, there are visible precipitations of Ni_2_Si at the boundaries of the phase α Figure 5, but also a few small Re phases visible on Figure 6.

The calculated values of heat capacity in the liquid state Cp_l_ and values of heat capacity in the solid state Cp_s_ for Cu–2Ni–1Si alloy before and after modification, taking into account the change in cooling rate, determined changes of crystallization heat of the structural elements, as shown in Table 3 and Table 4.

The Re phase solidifies as a board of the size of a few μm in the vicinity of phase with Ni and Si as shown in Figure 6a and confirmed by analysis of Figure 6b.

As a result of supersaturation, plastic deformation and subsequent ageing of Cu–2Ni–1Si–0.6Re alloys, multiple, very small rhenium precipitations of 200 to 600 nm in size were formed (Figure 7), which strengthen the alloy. At the same time, the precipitations of the Re phase causes the fragmentation of the microstructure and precipitation strengthening (Figure 7 and Figure 8). This causes a significant increase of hardness of the alloy after the addition of Re, and at the same time the alloy reveals a similar value of electrical conductivity to Cu–2Ni–1Si as shown in Table 4.

EDS analysis in the microareas has confirmed that small participates presented on the Figure 9a (Table 5) are the phases containing rhenium.

The increase in hardness is caused by precipitation of small, matrix-incoherent Re phases strengthening the alloy, but causing on the other side small decrease in conductivity compared to the alloy modified with Re (Table 6). However the increase of conductivity of the alloy after aging is caused by precipitation of intermetallic phases in the solid solution.

As a result of the use of heat and plastic treatment, the number of point and line construction defects increases. This facilitates diffusion and the formation of nanometric phases with Re (Figure 8 and Figure 10a). Figure 8 not only confirms the presence of nanometric Re phases but also shows their location and dimensions.

Tests using X-ray scanning transmission electron microscopy (STEM) with the application of chemical composition analysis with an EDS detector confirm the occurrence of Re phase precipitation in the Cu_α_ matrix. These studies also allowed us to state that Re is not dissolved in the Cu_α_ matrix. (Figure 10b).

TEM investigations allowed us to identify often very differentiated segregation zones of a defective network structure—with different dislocation density (Figure 11b).

Plastic treatment influences the formation of point and linear defects. It makes the diffusion of rhenium and phases’ formation easier, as shown in Figure 11b. Re addition also affects the blocking of recrystallization (Figure 11a).

The results based of the microstructure investigation using a high-resolution transmission electron microscope have revealed that the microstructure of the material shows a different structure revealing Rhenium particles in the range of ca. 200 to 500 nm, and the shape of the detected Rhenium particles was mainly round, without sharp edges. Based on TEM investigations, it was found that the size of the Re grain was confirmed on the atomic scale using a high-resolution transmission electron microscope (HRTEM). The electron diffraction results obtained allowed us to confirm rhenium as the particle phase with zone axis of $\left[1\bar{1}1\right]$ for each of the measured particles (Figure 12).

## 4. Conclusions

The following conclusions were drawn as a result of the investigations of rhenium-modified Cu–2Ni–1Si alloys performed:
The addition of rhenium in the form of 50% Ni–Re master alloy did not change the shape of the derivative curve during the crystallization of the alloy. The alloy was cooled from 1150 °C in order not to cause any overheating of the alloy. The alloy was annealed at the target temperature for 300 s. An increase in grain size was observed as a result of the tests due to a decrease in the crystallization rate in the range from *T_L_* to *T_SOL_* by 0.5 °C. This difference caused a two-fold increase in grain size. In the alloy with the addition of rhenium, the phases with Ni and Si have solidified at the grain boundary of the α phase, whereas, the Re phase has solidified in the form of plates (Figure 6a).As a result of the heat treatment consisting of solution heat treatment (SHT) and aging as well as plastic deformation, the Re phase was obtained with a size from 200 nm to 600 nm, The particles themselves are incoherent with the matrix and are responsible for strengthening the alloy. Diffraction investigations using high-resolution TEM identified nanometric Re phases with a zone axis of $\left[1\bar{1}1\right]$. As a result of using Re with a concentration of 0.6% by mass, the hardness increased by 50%, while the conductivity in the modified alloy remained at a comparable level.

## Figures and Tables

**Figure 1 materials-13-01182-f001:**
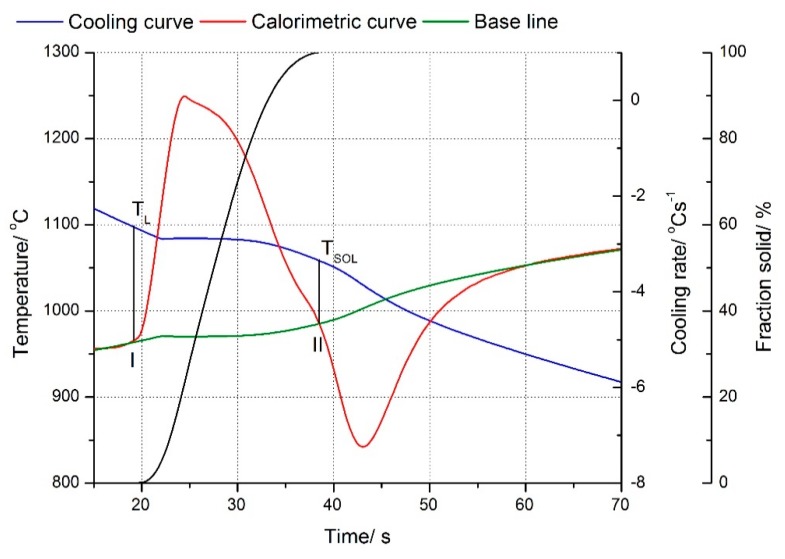
Cooling and derivative curves for Cu–2Ni–1Si alloys. Solid fraction during the crystallization of the particular compounds of Cu–2Ni–1Si alloy. Freely cooled alloy, cooling rate 2.2 °C s^−1^.

**Figure 2 materials-13-01182-f002:**
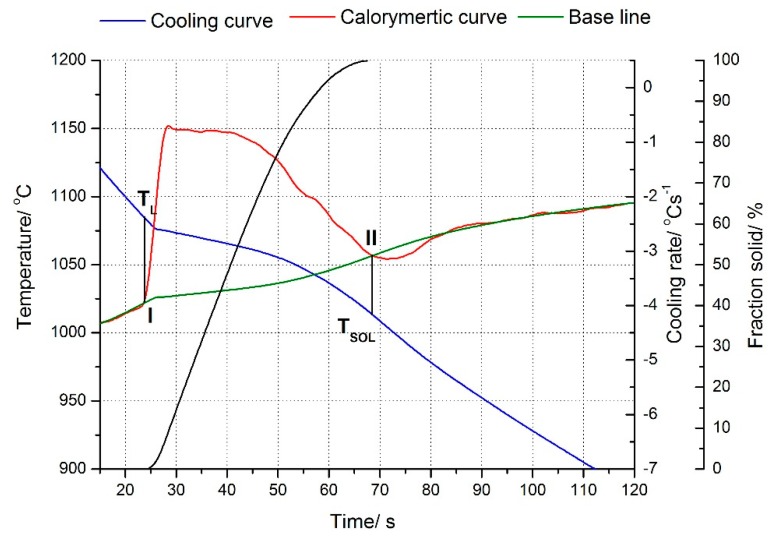
Cooling and derivative curves for Cu–2Ni–1Si–0.6Re alloys. Solid fraction during the crystallization of the particular compounds of Cu–2Ni–1Si–0.6Re alloy. Freely cooled alloy, cooling rate 1.5 °C s^−1^.

**Figure 3 materials-13-01182-f003:**
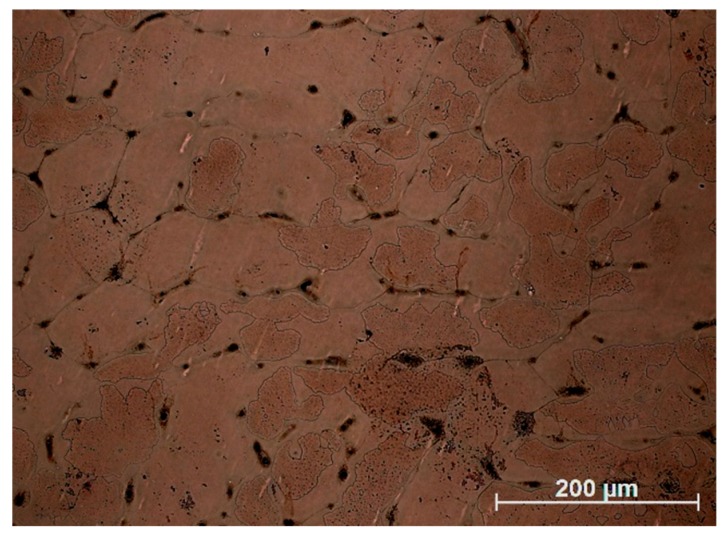
Microstructure of Cu–2Ni–1Si, initial state. Cooling rate 2.2 °C s^−1^.

**Figure 4 materials-13-01182-f004:**
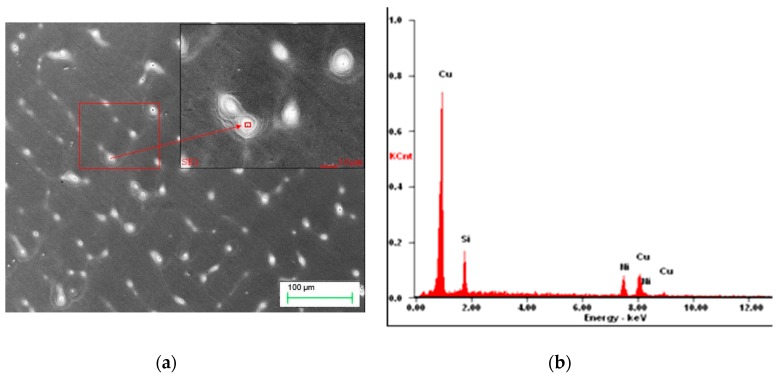
Microstructure of Cu–2Ni–1Si, initial state. Cooling rate 2.2 °C s^−1^ (**a**), energy-dispersive X-ray spectroscopy (EDS) analysis in microarea, mass concentration, %: Si 9.09; Ni 30.55; Cu 60.35 (**b**).

**Figure 5 materials-13-01182-f005:**
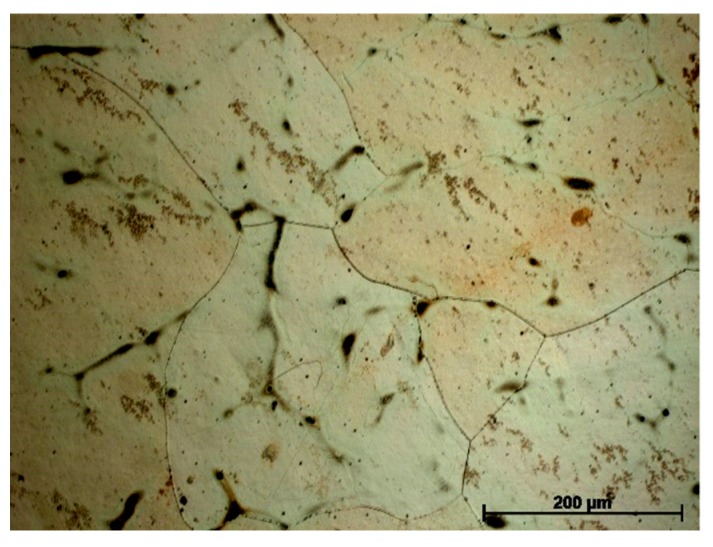
Microstructure of Cu–2Ni–1Si–0.6Re, initial state. Cooling rate 1.5 °C s^−1^.

**Figure 6 materials-13-01182-f006:**
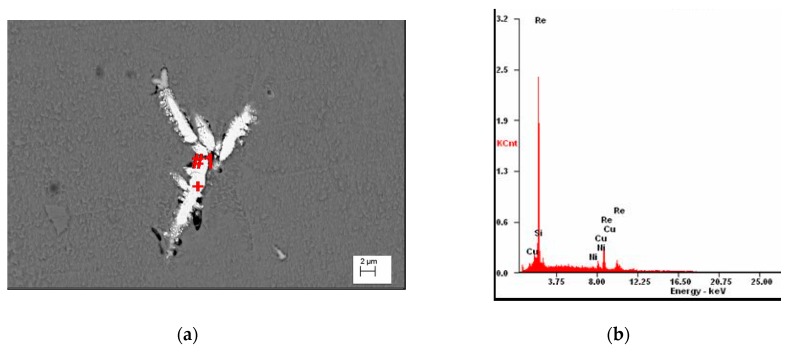
Microstructure of Cu–2Ni–1Si-0.6Re, initial state (**a**), EDS analysis in point #1 of elements mass concentration, %: Si 3.22; Re 86.1; Ni 1.9; Cu 8.78 (**b**).

**Figure 7 materials-13-01182-f007:**
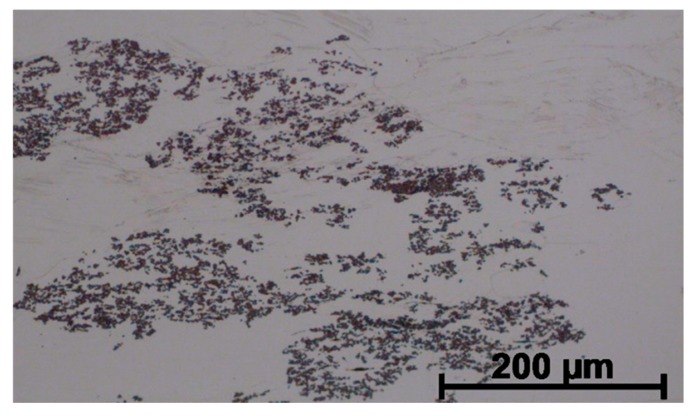
Structure of Cu–2Ni–1Si–0.6Re alloy: supersaturation 950 °C; time 1 h, water cooling (Ar); plastic deformation 50%; ageing 450 °C 1 h.

**Figure 8 materials-13-01182-f008:**
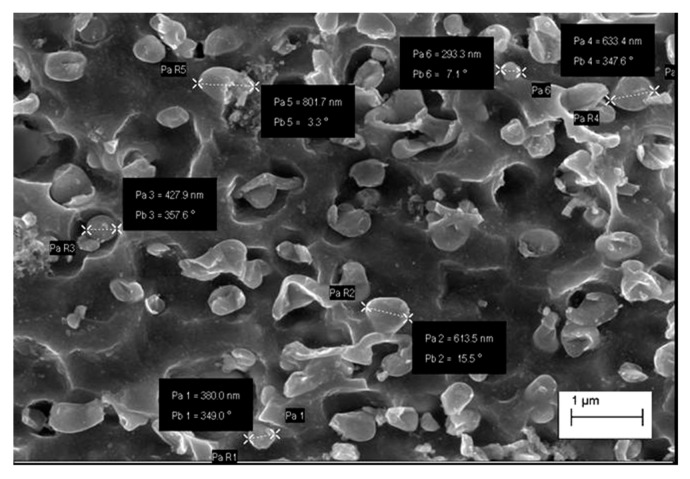
Structure of Cu–2Ni–1Si–0.6Re alloy: supersaturation 950 °C; time 1 h, water cooling (Ar); plastic deformation 50%; ageing 450 °C 1 h.

**Figure 9 materials-13-01182-f009:**
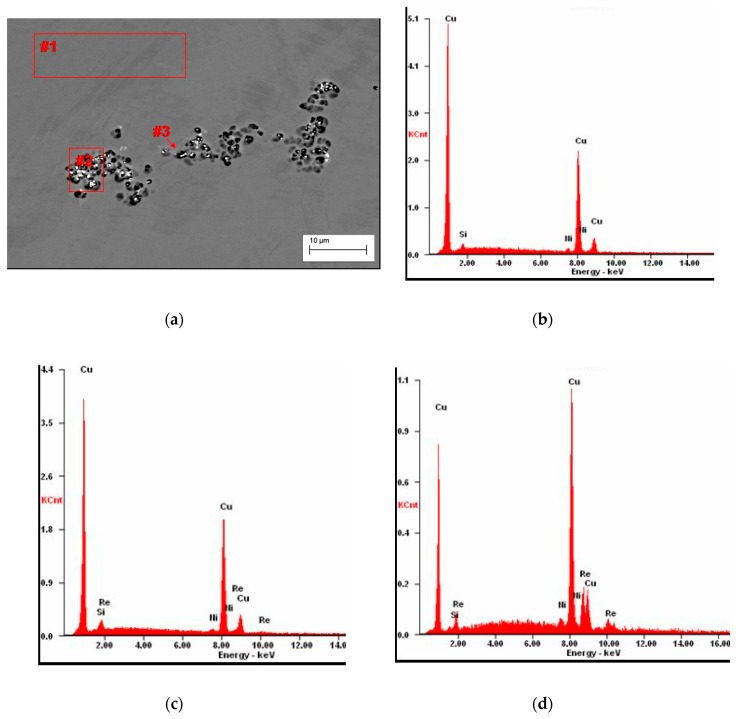
Structure of Cu–2Ni–1Si–0.6Re alloy: supersaturation 950 °C; time 1 h, water cooling (Ar); plastic deformation 50%; ageing 450 °C 1 h (Table 5) (**a**), analysis of EDS from the #1 (**b**), analysis of EDS from the microarea #2 (**c**), analysis of EDS from the microarea #3 (**d**).

**Figure 10 materials-13-01182-f010:**
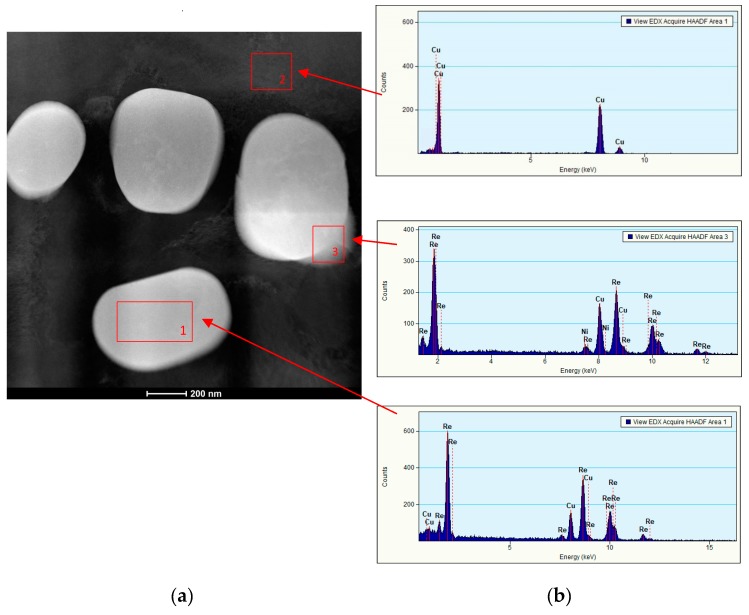
Structure of Cu–2Ni–1Si alloy modified with Re after heat treatment and cold plastic deformation (**a**), analysis of 1, 2 and 3 zones made with energy-dispersive X-ray spectroscopy (**b**).

**Figure 11 materials-13-01182-f011:**
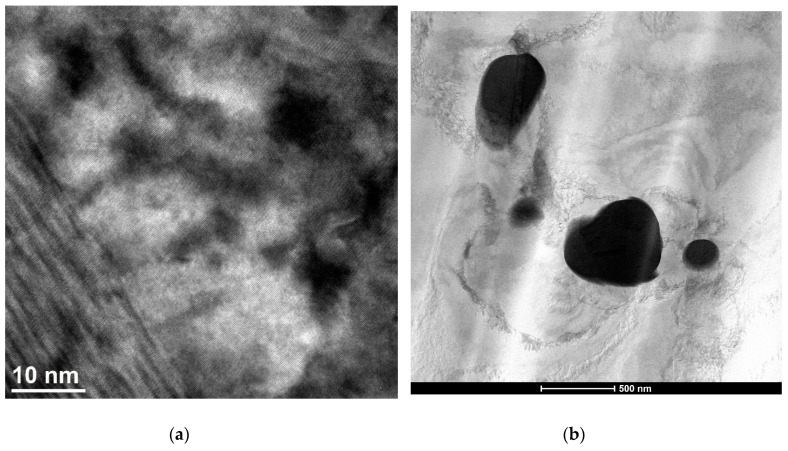
Transmission electron microscope (TEM) structure of Cu–2Ni–1Si–0.6Re alloy, supersaturation 950 °C; time 1 h, water cooling; plastic deformation 50%; ageing 450 °C 1 h (**a**), Re phase (**b**).

**Figure 12 materials-13-01182-f012:**
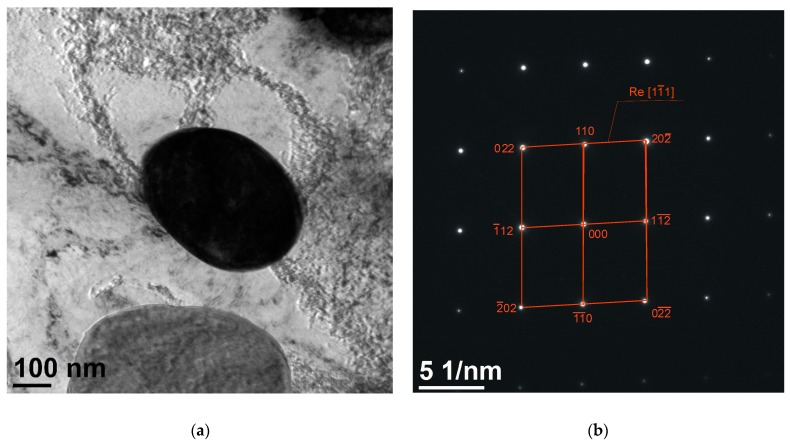
Structure of Cu–2Ni–1Si alloy modified with Re: dark field image (**a**), diffraction pattern of the zone axis for [1,2,3,4,5,6,7,8,9,10,11] Re (**b**).

**Table 1 materials-13-01182-t001:** Chemical composition of analyzed copper alloys.

Analyzed Alloy	Mass Concentration of the Elements in % Mass			
	Ni	Si	Re	Cu
**Cu–2Ni–1Si**	2	1	−	rest
**Cu–2Ni–1Si–0.6Re**	2	1	up to 0.6	rest

**Table 2 materials-13-01182-t002:** Results from thermal-derivative analyses of Cu–Ni–Si–Cr and Cu–Ni–Si–Cr–Re alloys.

Analyzed Alloy	Temperature, °C		Sample Mass, g
	T_L_	T_SOL_	
Cu–2Ni–1Si	1098	1057	11.36
Cu–2Ni–1Si–0.6Re	1083	1014	24.2

**Table 3 materials-13-01182-t003:** Latent heat of crystallization produced by phases and its percentage in the total heat of crystallization of Cu–2Ni–1Si alloy.

**Cu–2Ni–1Si**			
**Heat Capacity in Liquid State Cp_l_, J/g °C**	**Heat Capacity in Solid State Cp_s_, J/g °C**		**Weight of Sample, g**
0.448	0.392		11.36
**Reaction**	**Latent heat of crystallization**		**Percentage, %**
	**Samples, J**	**Unit Weight of a Sample, J/g**	
L → α	1295.31	114.02	98.01
L → α + Ni + Si	22.35	1.97	1.99
Total	1317.66	115.99	100

**Table 4 materials-13-01182-t004:** Latent heat of crystallization produced by phases and its percentage in the total heat of crystallization of Cu–2Ni–1Si–0.6Re alloy.

**Cu–2Ni–1Si–0.6Re**			
**Heat Capacity in Liquid State Cp_l_, J/g °C**	**Heat Capacity in Solid State Cp_s_, J/g °C**		**Weight of Sample, g**
0.460	0.364		24.2
**Reaction**	**Latent Heat of Crystallization**		**Percentage, %**
	**Samples J**	**Unit Weight of a Sample, J/g**	
L → α	4139.26	171.04	91.8
L → α + Ni + Si	305.84	12.64	8.2
Total	4445.10	183.68	100

**Table 5 materials-13-01182-t005:** Results of the EDS spectrum analysis for the areas from Figure 9 (wt.%).

Element	Area #1	Area #2	Point #3
**Si**	1.75	1.21	0.26
**Ni**	2.47	2.28	2.2
**Re**	−	12.38	36.19
**Cu**	95.78	84.13	61.35

**Table 6 materials-13-01182-t006:** Results of tests of microhardness (HV) and electrical conductivity (γ) of Cu–2Ni–1Si alloys in cast state and after heat and plastic treatment and of Cu–2Ni–1Si–0.6Re alloy in cast state and after heat and plastic treatment.

Alloy Symbol	Microhardness, HV	Conductivity, MS/m
Cu–2Ni–1Si	60	8
Cu–2Ni–1Si_HTPD *	150	14
Cu–2Ni–1Si–0.6Re	96	9
Cu–2Ni–1Si–0.6Re_HTPD *	225	12

HTPD *—after heat treatment and plastic deformation.

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
