# Peer review of "Precipitation Strengthening of Cu–Ni–Si Alloy"

_materials, 2020, doi:10.3390/ma13051182_

Round 1

Reviewer 1 Report

Abstract:

Rewrite the abstract part according to your results.

Introduction part: 

Please explain in detail the novelty and aim of this work.

Materials and Methods part:

Please change Cu-2Ni-1Si-0,6Re alloys  to Cu-2Ni-1Si-0.6Re.

Please mention the purity of this alloys

Please mention the microhardness test

Results part:

Please provide high quality SEM images in Figure 3 with EDS analysis data.

According to Figure 3 and Figure 4 the grain size of Cu-2Ni-1Si-0.6Re. is lagre than that of Cu-2Ni-1Si alloy. Please explain in details in discussion part.

Please provide EDS data in table in Figure 7.

Please add other conditions images in Figure 9.

Conclusions:

Please mention the grain size of all alloys

Author Response

Dear Reviewer 

I have included the answers to the review in a pdf file

Reviewer 2 Report

Manuscript ID: Materials-725480

Title:  Precipitation strengthening of Cu-Ni-Si alloy

Authors:  Beata Krupińska, Zbigniew Rdzawski, Mariusz Krupiński and Wojciech Pakieła

The manuscript deals with the experimental investigation of the influence of Rhenium over the strengthening mechanism and crystallization process of Cu-Ni-Si alloy. Also, the manuscript intend to reveals how the addition of Rhenium may affect mechanical/ electrical properties of alloys studied.

I appreciate that the proposed topic is interesting and useful and deserves to be studied.

In my opinion, the results included in manuscript respond partially to the assumed objectives. Moreover, the discussion paragraph should be substantially improved with in-depth insight on strengthening mechanism and crystallization process of Cu-Ni-Si alloy.

There is a lack of details about the graphs and pictures shown, especially referring to the morphology and correlation aspects.

Therefore, I would recommend to the authors to enrich the discussion of results with significant statements  about the findings of the research. I would suggest to be added supplementary results, if needed, in order to improve the manuscript. Consequently, the conclusions paragraph should be completed.

Also, I think the manuscript title does not completely reflect the work content.

Finally, I would recommend a careful proofreading of the text.

Author Response

(The authors gave the same response as above.)

Round 2

Reviewer 1 Report

Authors addressed all of my comments in revised manuscript.

Author Response

Dear Reviewer
Thank you very much for the re-review of my article.

Kind regards,
Beata Krupińska

Reviewer 2 Report

The authors have added some phrases aiming to enrich the abstract, introduction, methodology, results and discussions and conclusions. I noticed the addition of new data in tables and also new graphs (Figs. 10, 11 and 12). Unfortunately, the manuscript still suffers because of a lack of comments related to the abundant graphical and tabular data; the new Figures 10, 11 and 12 are not discussed at all.

The quality of the Figures 5, 6 and 8, that contain materials spectra, is poor which makes them impossible to be understood.

In my opinion, the manuscript must be improved, especially with consistent comments over the results and data obtained. Also, phenomenological explanation and further correlation with the results have to be done. Similarly, the conclusions should be improved.

I noticed editing errors, and therefore I recommend a carefully review of the text.

Author Response

Dear Reviewer

Thank you very much for the re-review of my article. Thank you for the comments made while reading my article.

Round 3

Reviewer 2 Report

I took note of the corrections and the improvement of the manuscript, in relation to the recommendations of the reviewer. Also, a further verification of the text was made. So, I appreciate that the manuscript can be considered for publication in the journal Materials, in its current form.